# Associations between community health workers' home visits and education-based inequalities in institutional delivery and perinatal mortality in rural Uttar Pradesh, India: a cross-sectional study

Andrea Katryn Blanchard [ID],[1] Tim Colbourn,[2] Audrey Prost,[2] Banadakoppa Manjappa Ramesh,[1] Shajy Isac,[1,3] John Anthony,[1,3] Bidyadhar Dehury,[3] Tanja A J Houweling[4]

For numbered affiliations see end of article.

**Correspondence to**
Dr Andrea Katryn Blanchard; andrea.blanchard@umanitoba.ca

## ABSTRACT

**Introduction** India's National Health Mission has trained community health workers called Accredited Social Health Activists (ASHAs) to visit and counsel women before and after birth. Little is known about the extent to which exposure to ASHAs' home visits has reduced perinatal health inequalities as intended. This study aimed to examine whether ASHAs' third trimester home visits may have contributed to equitable improvements in institutional delivery and reductions in perinatal mortality rates (PMRs) between women with varying education levels in Uttar Pradesh (UP) state, India.

**Methods** Cross-sectional survey data were collected from a representative sample of 52 615 women who gave birth in the preceding 2 months in rural areas of 25 districts of UP in 2014–2015. We analysed the data using generalised linear modelling to examine the associations between exposure to home visits and education-based inequalities in institutional delivery and PMRs.

**Results** Third trimester home visits were associated with higher institutional delivery rates, in particular public facility delivery rates (adjusted risk ratio (aRR) 1.32, 95% CI 1.30 to 1.34), and to a lesser extent private facility delivery rates (aRR 1.09, 95% CI 1.04 to 1.13), after adjusting for confounders. Associations were stronger among women with lower education levels. Having no compared with any third trimester home visits was associated with higher perinatal mortality (aRR 1.18, 95% CI 1.09 to 1.28). Having any versus no visits was more highly associated with lower perinatal mortality among women with lower education levels than those with the most education, and most notably among public facility births.

**Conclusions** The results suggest that ASHAs' home visits in the third trimester contributed to equitable improvements in institutional deliveries and lower PMRs, particularly within the public sector. Broader strategies must reinforce the role of ASHAs' home visits in reaching the sustainable development goals of improving maternal and newborn health and leaving no one behind.

## Strengths and limitations of this study

► This study analysed a large population-based survey that allowed the detection of education-based inequalities in institutional delivery and perinatal mortality rates (PMRs) overall and at public and private health facilities separately, as well as analyses of whether these inequalities were lower among women who received community health workers' home visits in the third trimester.

► The survey included women who gave birth within the last 60 days, which likely improved their recall and classification of perinatal compared with late neonatal deaths.

► The cross-sectional survey design precluded conclusions about causality, and the results are not directly generalisable to non-high priority districts in Uttar Pradesh or other states in India.

► The first round of the survey did not include wealth or asset indices, which would have enabled a broader exploration of inequalities in institutional delivery and PMRs.

► Survey responses were self-reported, which could lead to recall, social desirability or misclassification biases.

## BACKGROUND

Since the establishment of the sustainable development goals, global health leaders have renewed their pledge to improve maternal and newborn health, with an emphasis on reducing high mortality rates among families who are most socioeconomically disadvantaged.[1–3] In 2015, 2.7 million infants died in the first month of life, accounting for 45% of child deaths globally. A further 2.6 million were estimated to be stillborn in 2015.[4] Despite considerable progress in the past

decades, over 1 million neonatal deaths and 1 million stillbirths are estimated to occur every year in South Asia.[5] About a quarter of neonatal deaths globally are estimated to occur in India alone.[6] The greatest number of deaths happen during the perinatal period, that is, between 28 completed weeks of gestation and 7 days after birth.[7 8] Uttar Pradesh (UP) is India's most populous state and has had the highest perinatal mortality rate (PMR), estimated at 56 per 1000 live births between 2010 and 2015, compared with 36 per 1000 live births nationally.[9] Uptake of health services for Reproductive, Maternal, Newborn and Child Health (RMNCH) has historically been low and inequitable between socioeconomic groups in many regions of UP. Nearly 68% of women in UP reported delivering in a health facility in 2015/2016. This ranged from 56% among women with no education to over 80% for those with 10 or more standards of education overall, while the educational differences were greater at private than public facilities.[10 11]

Community-level interventions are a promising approach to equitably improve maternal and newborn health; most of these rely on community health workers (CHWs).[12–14] CHWs have been defined as people who live and work in their communities, are selected by and accountable to them, and may or may not be part of the health system.[15–17] The pursuit of universal health coverage has given CHW programmes renewed importance in public health programmes.[3 18 19] Global health leaders have endorsed their potential to improve institutional delivery in resource-constrained settings.[14 20 21] There is also evidence that CHWs can promote safe newborn care practices that prevent perinatal mortality.[22–26]

India's National Health Mission (NHM) has recruited and trained a large cadre of CHWs called Accredited Social Health Activists (ASHAs) since 2005. ASHAs are trained to keep a birth register of all pregnant women as well as routinely contact and counsel women and their families on antenatal, delivery, and postnatal care through home visits. During home visits in the third trimester of pregnancy, ASHAs convey the benefits of institutional delivery, review birth plans and counsel families on essential newborn care practices, all of which are intended to reduce perinatal mortality.[27] ASHAs and the women they counsel normally receive monetary incentives through the *Janani Suraksha Yojana* scheme for giving birth at a health facility. Services are meant to be free at public health facilities. ASHAs should normally inform women about these services and monetary incentives when preparing a birth plan. They also receive training on how to better reach geographically and socioeconomically marginalised families.[28] Despite the scale of the ASHA home visiting programme, few studies have examined whether it has improved neonatal health outcomes equitably (as much or more among lower compared with higher socioeconomic position groups).[29–31] There is a paucity of research to examine its effects on equity in perinatal mortality overall or comparing between public and private facility births within India's increasingly pluralistic health system.

Our study's overall aim was to examine whether ASHAs' home visits in the third trimester of pregnancy were associated with equitable improvements in institutional delivery and reductions in perinatal mortality rates at public and private facilities between women with lower to higher education levels. The study had three related research questions:

1. Do women with higher compared with lower education levels report (a) higher institutional delivery and (b) lower PMRs, overall and by place of birth (home, public, and private facilities)?
2. Is women's exposure to any third trimester home visits associated with (a) higher institutional delivery and (b) lower PMRs, overall and by place of birth?
3. Is exposure to third trimester home visits more strongly associated with (a) higher institutional delivery and (b) lower PMRs, for women with lower compared with higher education levels overall and by place of birth?

The conceptual models that guided the study's analyses are shown in online supplemental figures 1 and 2. This study was part of a larger mixed-methods enquiry into the role of ASHAs' home visits in reducing inequities in perinatal health between women of higher and lower socioeconomic positions in 25 districts of rural UP state, India.

## METHODS

### Study design and setting

The government of UP's branch of the NHM has committed substantial resources towards reducing the burden of maternal and neonatal mortality. In 2013, UP's NHM programme established a partnership with the India Health Action Trust and the University of Manitoba in Canada, under the auspices of the UP Technical Support Unit, funded by the Bill and Melinda Gates Foundation. This partnership was embedded within UP's NHM programmes in its 25 high priority districts (HPDs), covering a population of over 83 million people.[32] The government chose the HPDs by ranking districts using a composite health index that combined six health indicators from the state's Annual Health Survey data.[33] This study used data from the first round of the UP Technical Support Unit's community-based, cross-sectional Community Behavioural Tracking Survey (CBTS-1) collected in April 2014 to February 2015, which included 100 blocks (sub-districts) of the 25 HPDs.

### Sampling and participants

Sampling was done in two stages, starting with randomly selecting the primary sampling units (PSU), and then inviting all eligible women within them to participate. For the first stage, the survey's PSU was an ASHA area, which covers approximately 1000 people and is the smallest unit of health service delivery. The required sample size in each block to detect a minimum percentage change of 7.5% in the institutional delivery rate within 6 months

(a key indicator expected to change between survey rounds), assuming an estimated non-response rate of 15% and a design effect of 1.5, was 536 women per block or 53 600 across the 100 blocks. This was considered an ideal sample size given the resources for data collection, compared with what was required for a higher or lower minimum detectable percentage change in institutional delivery rate (5% would give n=1209; 7.5%, n=536; 10%, n=300). The average number of PSUs needed to reach a sample size of at least 536 participants per block was estimated to be 110, assuming that there would be five eligible respondents who had completed a pregnancy within the last 60 days per PSU. This assumption was based on the crude birth rate of UP in the most recent available data (29 per 1000 population in the Annual Health Survey in 2011/2012). After all PSUs within the 100 blocks of 25 districts were listed, 110 PSUs were chosen in each block (11 000 ASHA areas in total) using systematic random sampling.

For the second stage, every household was enumerated and all eligible women in the household were invited to participate in each of the randomly selected 11 000 PSUs. PSUs were of nearly uniform size and so did not require sample weighting. Eligible participants were women aged 15–49 who had completed a pregnancy in the past sixty days. In total, 72 054 women were identified and 57 788 were included in the survey, leading to an 80% response rate. There were no refusals to participate; non-response occurred largely when women were repeatedly unavailable at home when the interviewers visited on different days. In addition, 5173 participants responded that their pregnancy ended in an abortion, and so were not included in this study's sample. We analysed data from 52 615 participants. For analyses on perinatal mortality, the sample used was 52 588: 27 cases (0.05%) could not be included as they had invalid responses. A flow diagram of the sampling process is shown in online supplemental figure 3.

## Data collection

Field interviewers' educational background included a minimum of a Bachelor degree in social sciences or related field, and 6 months or more of related experience. They received 10 days of intensive training on data collection methods, including orientation and then hands-on field practice with input from their supervisors. To collect the data, the interviewers first determined the PSU boundary. With a random start, they visited all households and gave each household a number. They used a screening questionnaire in each household to identify women who recently completed a pregnancy, and then described the study to them and asked for their informed consent. Interviews were conducted in Hindi, the local language. Data were collected anonymously using a mobile application, and routinely checked by field supervisors. The survey tool included close-ended questions related to selected background characteristics, pregnancy outcomes, antenatal care, birth preparedness, delivery, postnatal and newborn care, and reproductive health.

## Variables

### Outcome variables

The outcomes of interest were institutional delivery and PMRs. We conducted the analyses with the binary outcome of institutional delivery (public and private) versus home, as well as for institutional delivery at a public facility versus home and a private facility versus home separately. PMR was calculated as the number of stillbirths (occurring between 28 weeks of gestation and delivery) and deaths during the early neonatal period (first 7 days of life) out of 1000 births. These were self-reported and not confirmed by verbal autopsies.

### Independent variables

Equity in health has been defined as the absence of systematic disparities in health or its social determinants between groups with differing positions in a social hierarchy.[34] A number of socioeconomic position characteristics are used to describe inequities in health, in terms of wealth or asset indices, education, occupation, caste, ethnicity and religion.[34 35] Wealth or asset-based inequalities would have been valuable to assess, but this indicator was not available in the first round of the CBTS. We examined caste group differences, but they were more consistent for institutional delivery rates than PMR. We focused on education as the main indicator of socioeconomic position in this study as there were consistent inequalities in both outcomes. Higher maternal education has been associated with child survival and utilisation of maternal health services in previous research.[36] Education levels represent socioeconomic opportunities that are achieved early in life and remain fairly stable over the lifetime, being relatively uninfluenced by later health status. It is also an ordinal variable and enables the examination of gradients in health outcomes.[37] We categorised education levels into four groups from least to most years of study (standards) based on common Indian school system divisions: none or illiterate, 1–5 (lower primary school), 6–10 (higher primary to lower secondary) or more than 10 standards (upper secondary and college).[37]

Receipt of ASHA home visits in the third trimester of pregnancy was the main intervention exposure variable. We considered exposure to home visits earlier in pregnancy, but only 9.8% of women received a home visit in the first two trimesters of pregnancy and not in the third trimester in the CBTS-1. This suggested that the binary variable of any third trimester home visit was a good single indicator of ASHA home visit coverage. Third trimester visits are also meant to focus substantially on encouraging and preparing women for institutional delivery.

### Covariate and confounding variables

Women's age, caste, religion and parity are known to be associated with place of birth, as well as their risk of

perinatal loss; all but parity were available in the CBTS-1 data.[34–36 38–40] Having any antenatal care visits can increase the likelihood of birth preparedness and relatedly, having an institutional delivery as well.[36 38] Pregnancy, intra partum and newborn complications have also been found to lead to higher institutional delivery rates, and all can affect mortality rates.[38] Among these, the available covariates in the dataset that were found to be significant at p<0.05 in bivariable models were retained in multivariable models.

Maternal age and caste (ordinal categorical variables) were adjusted for as confounders in all multivariable models. Religion and any antenatal check-ups (binary variables) were included for institutional delivery, but not PMR, because they were significantly associated with the former but not the latter in the bivariable models. We also adjusted for the binary variables of experiencing any pregnancy complications in analyses with both outcomes, as well as any intrapartum complications, and then place of delivery for perinatal mortality. Pregnancy complications were defined as experiencing any of the following: excessive bleeding before delivery, convulsions, high blood pressure or sepsis/fever. Intrapartum complications included any of the following: premature labour, preterm/premature rupture of the membrane, prolonged labour of more than 12 hours, obstructed labour, breach/malpresentation, or excessive bleeding immediately after delivery.

## Statistical analyses

Data management and analyses were conducted in STATA V.13.0. First, we described the independent variables, outcomes and covariates, using tabulations to produce proportions and related confidence intervals (CIs) for all variables; for PMRs, we summarised the mean number of women reporting a perinatal death with CIs, and calculated the rate of perinatal deaths per 1000 births.

Our first question was whether women of higher compared with lower education levels reported (a) higher institutional delivery and (b) lower PMRs, overall and by place of birth. We used cross-tabulations to describe the proportion of women reporting an institutional delivery, in each education group overall and at each place of delivery. Similarly, we calculated the PMR among women in each education group, overall and by place of birth.

To answer our second question, we aimed to assess if women's exposure to any third trimester home visits was associated with (a) higher institutional delivery and (b) lower PMR, overall and by place of birth. We used bivariable and multivariable generalised linear models to assess whether (a) institutional delivery and then (b) PMR, were independently associated with having any third trimester home visits, before and after adjusting for the covariate and confounding factors. Each of the models were run first for all births, and then stratified to compare home, public and private facility births.

For the final question, we examined whether exposure to third trimester home visits was more strongly associated with (a) higher institutional delivery and (b) lower perinatal mortality for women with lower compared with higher education levels, overall and by place of birth. We descriptively assessed whether there was higher institutional delivery and lower PMR, among women with lower compared with higher education if they had any compared with no third trimester home visits, by place of delivery. We then used stratified unadjusted and adjusted generalised linear models to examine whether (a) higher institutional delivery and (b) lower PMR outcomes, were more strongly associated with women's exposure to third trimester home visits for women with lower compared with higher education groups, overall and by place of delivery. Interaction terms between exposure to home visits and the outcomes were also initially examined. These were significant for institutional delivery in the adjusted models, but not for PMR; the latter showed uneven gradients by education groups that were likely obscured in the interaction term. We therefore focused on the stratified results as they were more interpretable and transparent about where uneven patterns of inequalities and associations existed. Sensitivity analyses were not applicable for the present analysis.

In the analyses to address analytical questions 2 and 3, multivariable models with institutional delivery as outcome were adjusted for maternal age, caste, religion, any pregnancy complications and any antenatal check-ups. The multivariable models with PMR as the outcome were all adjusted for maternal age, caste, any pregnancy complications and any intrapartum complications. After adjusting for these covariates, the models with PMR were also adjusted for place of birth to assess how this affected the relationship between exposure to home visits and PMR, before stratifying by place of birth. Ordinal variables were indicated as such in the STATA commands to obtain distinct estimates for each category relative to a reference category.

The generalised linear models used the binomial family distribution and a log link function to obtain risk ratios (RR). Some adjusted models with binomial distribution failed to converge for the institutional delivery outcome. Others have found that this can occur when including variables with few data points in some categories, and addressed this by using a Poisson distribution with robust standard errors to produce the RR.[41] Using robust Poisson models with a log link function produced estimates that were almost identical to our analyses using a log binomial distribution in models that converged. Therefore, the RR was produced from robust Poisson and log binomial models for institutional delivery and PMR, respectively. For both outcomes, we also present the unadjusted and adjusted percentages of institutional delivery and PMRs per 1000 live births, alongside the unadjusted and adjusted RR and CI estimates in the results. To aid interpretation, results for all three research questions are

**Table 1** Descriptive sociodemographic and health-related characteristics of the sample

| Variable | Total, n (%) | Any third trimester home visit (%) | Institutional delivery | | | Perinatal mortality rate (per 1000 births)* |
| | | | Total (%) | Public facility (%) | Private facility (%) | |
| --- | --- | --- | --- | --- | --- | --- |
| Overall | 52 615 (100) | 55 | 63 | 52 | 11 | 45 |
| Age | | | | | | |
| 15–24 years | 21 079 (40) | 55 | 70 | 56 | 14 | 49 |
| 25–29 years | 21 400 (41) | 55 | 62 | 51 | 11 | 39 |
| 30+ years | 10 136 (19) | 53 | 55 | 47 | 8 | 49 |
| Religion | | | | | | |
| Hindu | 42 720 (81) | 55 | 64 | 53 | 11 | 45 |
| Muslim/other | 9895 (19) | 51 | 59 | 46 | 13 | 42 |
| Caste | | | | | | |
| Scheduled tribe | 2216 (4) | 40 | 55 | 49 | 6 | 43 |
| Scheduled caste | 12 334 (23) | 55 | 62 | 54 | 8 | 44 |
| Other backward class | 24 579 (47) | 55 | 63 | 52 | 11 | 45 |
| General caste | 6323 (12) | 53 | 72 | 53 | 19 | 48 |
| Do not know | 7163 (14) | 57 | 61 | 49 | 12 | 44 |
| Education | | | | | | |
| None/illiterate | 30 614 (58) | 52 | 56 | 48 | 8 | 48 |
| 1–5 standards | 5693 (11) | 57 | 66 | 55 | 11 | 46 |
| 6–10 standards | 9895 (19) | 58 | 73 | 59 | 14 | 40 |
| >10 standards | 6413 (12) | 58 | 84 | 60 | 24 | 35 |
| Any pregnancy complications | | | | | | |
| No | 43 102 (82) | 55 | 63 | 52 | 11 | 38 |
| Yes | 9513 (18) | 51 | 67 | 53 | 14 | 76 |
| Any intrapartum complications | | | | | | |
| No | 32 000 (61) | 55 | 60 | 51 | 9 | 32 |
| Yes | 20 615 (39) | 53 | 69 | 54 | 15 | 65 |
| Any antenatal care check-ups | | | | | | |
| No | 27 179 (52) | 44 | 54 | 45 | 9 | 45 |
| Yes | 25 436 (48) | 66 | 73 | 59 | 14 | 45 |
| Any third trimester home visits | | | | | | |
| No | 23 914 (45) | NA | 54 | 42 | 12 | 50 |
| Yes | 28 701 (55) | NA | 71 | 61 | 10 | 41 |

*There were 27 missing values (0.05% of 52 615 responses) for the perinatal mortality variable, due to invalid responses.

presented first for institutional delivery, and then for perinatal mortality.

### Patient and public involvement statement

No patients or members of the public were involved in this study.

### RESULTS

Table 1 presents participants' characteristics. Eighty-one per cent of women were 15–29 years old (table 1). The majority of women belonged to other backward class groups (47%), followed by scheduled caste (23%) and general caste (12%) groups. Also, 58% of women were illiterate or had no formal education. Over half of the participants (55%) received any home visits during the third trimester, and 48% had gone for any antenatal care check-up. About 18% of women reported having any pregnancy complications, while 39% reported intrapartum complications.

### Institutional delivery

Almost two-thirds (63%) of women gave birth at a health facility (52% in a public facility, 11% in a private facility). To address research question 1(a), institutional delivery

was higher for women with the highest compared with lowest education levels (84% vs 56%, respectively). Educational inequalities were apparent for both public and private facilities (table 1).

Relating to the second question, a greater proportion of women who had any versus no third trimester home visits reported giving birth in a public facility (61% vs 42%, respectively). An equal proportion had private facility births with or without any third trimester home visits (10% vs 12%, respectively) (table 1). Women who had received any third trimester home visit compared with none reported higher institutional delivery overall (71% vs 54%) (unadjusted RR 1.31 (95% CI 1.29 to 1.33)) (online supplemental table 1). This association remained significant after adjusting for confounders (adjusted RR (aRR) 1.23 (95% CI 1.22 to 1.25)). The association between having any third trimester home visit and an institutional birth was stronger for births at a public facility (aRR 1.32 (95% CI 1.30 to 1.34)) than those at a private facility (aRR 1.09 (95% CI 1.04 to 1.13)) compared with home (table 2).

Turning to research question 3(a), figure 1 indicates that a higher proportion of women gave birth at public facilities versus home when they had any compared with no third trimester visits from an ASHA, and this was most noticeable among the least educated groups.

Similarly, the adjusted associations shown in table 2 between having an institutional birth (at both public and private facilities) and any third trimester home visit were stronger with each step down the educational gradient. Stratifying by facility type, this was more noticeable among public than private facility births compared with home.

### Perinatal mortality rate

The overall PMR in the CBTS-1 sample was 45 per 1000 births (table 1). The greatest number of early neonatal deaths occurred on the day of birth, and steadily decreased in number on subsequent days (online supplemental figure 4). The PMR was 39 per 1000 births among women giving birth at home, 43 per 1000 births at public facilities, and 73 per 1000 births at private facilities overall. For question 1(b), the PMR reduced with increasing educational attainment, from 48 to 35 per 1000 births among women with no education compared with more than 10 standards, respectively. Education inequalities in PMR were evident among home, public and particularly private facility births.

On question 2(b), the PMR was higher among women without compared to with any third trimester home visit, at 50 compared with 41 per 1000 births, respectively (unadjusted RR 1.21 (95% CI 1.12 to 1.31)) (online supplemental table 2). The association remained after adjustment for sociodemographic characteristics, any pregnancy and intrapartum complications (aRR 1.17 (95% CI 1.08 to 1.27)), and after additional adjustment for place of birth (aRR 1.18 (95% CI 1.09 to 1.28)) (table 3). Comparing between places of birth, the association between having no third trimester home visit and

PMR was strong for births at public facilities (aRR 1.31 (95% CI 1.17 to 1.47)). There was no association for births at private facilities (aRR 1.12 (95% CI 0.93 to 1.34)) or for home births (aRR 1.04 (95% CI 0.91 to 1.21)).

In relation to question 3(b), figure 2 descriptively suggests that for women who had any versus no third trimester home visits, educational differences in PMR were smaller among public facility births and to a lesser extent private facility births. The adjusted models showed that the association between having no third trimester home visits and higher PMR was stronger for those with one to five standards of education (aRR 1.53 (95% CI 1.20 to 1.94)), and to a lesser extent 5–10 standards (aRR 1.27 (95% CI 1.05 to 1.54)). The association appeared somewhat weaker for women with no education (aRR 1.13 (95% CI 1.02 to 1.25)), and absent among women who had an education over 10 standards (table 3).

When stratifying the adjusted models by place of birth, PMR was significantly higher among those without third trimester home visits compared with those with any visits for public facility births for all except the most educated group (table 3). However, differences in PMR between women with and without third trimester home visits were not significant for those who delivered at home or in private facilities, across all education groups. CIs were wide in some of the comparisons based on lower numbers of perinatal deaths.

## DISCUSSION

This study sought to examine the associations between women's exposure to ASHAs' third trimester home visits and education-based inequalities in institutional delivery and perinatal mortality rates in 25 districts of UP, India. The results indicate that there were education-based inequalities in institutional delivery and PMR, overall and at each delivery place. Women's receipt of any ASHA home visits during the third trimester of pregnancy was associated with giving birth in public facilities, and this association was increasingly strong for women with decreasing education levels. Women's receipt of any ASHA home visit in the third trimester was also significantly associated with lower PMR, and most strongly for women giving birth in public facilities among all but the most educated.

There were some strengths and limitations to the analyses. CBTS-1 survey responses were self-reported, and therefore prone to recall, misclassification and social desirability biases. The survey included women who delivered in the past 60 days to improve recall. However, the response rate was reduced somewhat because some women were not yet home from the facility or went to their maternal home in this period. The measure of an 'early' neonatal death could have been over-reported or under-reported, though combining stillbirths and early neonatal deaths to measure PMR can avoid the common misclassification of early neonatal deaths as stillbirths. We did not observe heaping of reported neonatal deaths

**Table 2** Associations between institutional delivery and any third trimester home visits, overall and by education level

| Third trimester home visits | All education groups (n=52615) | | None/illiterate (n=30614) | | 1–5 standards (n=5693) | | 6–10 standards (n=9895) | | >10 standards (n=6413) | |
|---|---|---|---|---|---|---|---|---|---|---|
| | Adjusted % (95% CI) | Adjusted RR (95% CI) | Adjusted % (95% CI) | Adjusted RR (95% CI) | Adjusted % (95% CI) | Adjusted RR (95% CI) | Adjusted % (95% CI) | Adjusted RR (95% CI) | Adjusted % (95% CI) | Adjusted RR (95% CI) |
| *Institution (public and private) vs home* | | | | | | | | | | |
| None (n=23914) | 54.2 (53.6 to 54.9) | Ref | 47.2 (46.4 to 48.0) | Ref | 57.5 (55.6 to 59.5) | Ref | 67.2 (65.7 to 68.6) | Ref | 81.6 (80.1 to 83.1) | Ref |
| Any (n=28701) | 71.1 (70.6 to 71.6) | 1.23 (1.22 to 1.25) | 62.9 (62.2 to 63.6) | 1.33 (1.30 to 1.36) | 71.3 (69.7 to 72.8) | 1.24 (1.19 to 1.29) | 77.3 (76.2 to 78.4) | 1.15 (1.12 to 1.18) | 85.6 (84.4 to 86.7) | 1.05 (1.03 to 1.07) |
| *Public facility vs home* | | | | | | | | | | |
| None (n=20997) | 47.9 (47.2 to 48.9) | Ref | 42.4 (41.5 to 43.2) | Ref | 51.5 (49.4 to 53.7) | Ref | 60.6 (58.9 to 72.9) | Ref | 74.1 (72.1 to 76.1) | Ref |
| Any (n=25682) | 67.7 (67.1 to 68.3) | 1.32 (1.30 to 1.34) | 59.9 (59.2 to 60.7) | 1.41 (1.38 to 1.45) | 68.4 (66.7 to 70.1) | 1.33 (1.26 to 1.39) | 74.1 (72.9 to 75.3) | 1.22 (1.18 to 1.26) | 81.9 (80.5 to 83.3) | 1.11 (1.07 to 1.14) |
| *Private facility vs home* | | | | | | | | | | |
| None (n=13860) | 22.6 (22.0 to 23.3) | Ref | 13.7 (13.0 to 14.4) | Ref | 22.8 (20.6 to 25.0) | Ref | 33.8 (31.8 to 35.9) | Ref | 61.4 (58.9 to 64.0) | Ref |
| Any (n=11318) | 24.6 (23.9 to 25.3) | 1.09 (1.04 to 1.13) | 16.6 (15.8 to 17.5) | 1.21 (1.12 to 1.31) | 23.9 (21.6 to 26.2) | 1.05 (0.91 to 1.20) | 35.1 (33.1 to 37.1) | 1.04 (0.95 to 1.13) | 58.0 (55.3 to 60.6) | 0.94 (0.87 to 1.00) |

CI, confidence interval; RR, risk ratio.

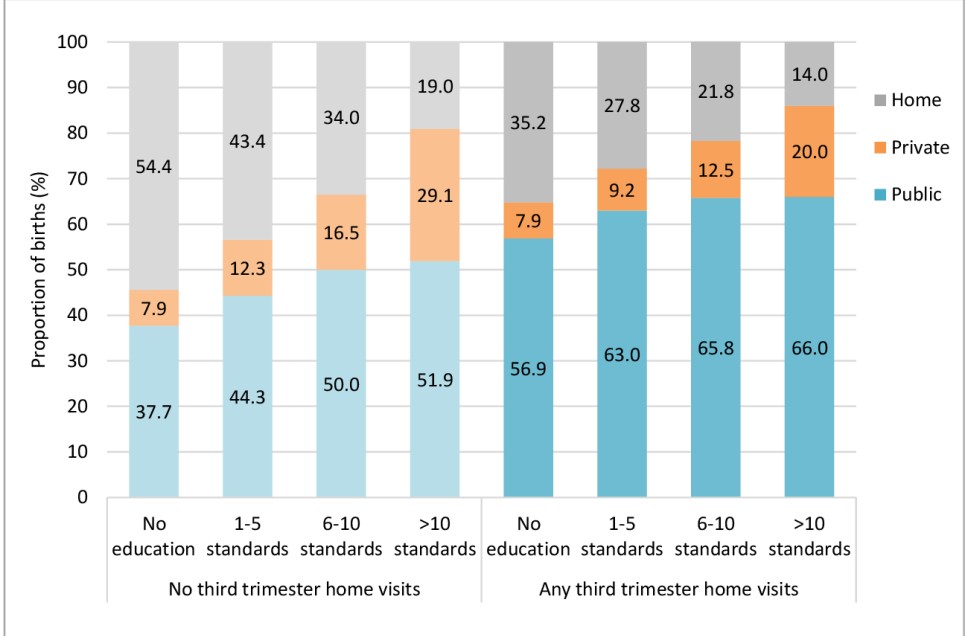

**Figure 1** Proportion of women in each education group giving birth at home, public facilities or private facilities, by exposure to any third trimester home visits.

within the first 7 days after birth, and the proportion of neonatal deaths at various times after delivery was similar across education groups. The CBTS-1 offered a sufficiently large population-based sample to measure PMR, and to stratify the analyses by education and place of delivery. However, its purpose as a programme monitoring methodology resulted in lack of data on some relevant factors or confounders, such as wealth, parity, mode of previous deliveries, newborn's sex, or having a singleton or twin birth that might have attenuated the strength of observed associations if included.[42] The cross-sectional nature of the analysis precluded conclusions about causality. Moreover, we assumed that the associations between exposure and outcome variables could involve reverse or multidirectional, rather than linear, pathways of causation.[43] For example, ASHAs may have more often chosen to visit women who preferred public facility births as a way to receive their incentives, while women with lower education may more often opt for the less expensive public services. The results may not be directly generalisable to non-HPD districts of UP, where institutional delivery was likely higher and PMR lower, and possibly less unequal.

Our results suggested that having any third trimester home visits from an ASHA was associated with a greater chance of women having an institutional delivery, and particularly at public facilities. Further, having any third trimester home visits was more greatly associated with having an institutional delivery for women with lower education levels. Other studies have reported positive associations between exposure to community-based interventions and lower socioeconomic inequalities in institutional delivery and skilled birth attendance in UP. Another cross-sectional study found that having any ASHA home visit was associated with higher institutional delivery

among poorer and illiterate women, but not among richer or more educated women.[30] A quasi-experimental study of an intervention involving NGO technical support to the ASHA programme in two districts of UP found that skilled birth attendance became more equitable between wealth groups in the intervention district but not the comparison district.[29] The role of ASHAs' home visits in increasing the proportion of institutional deliveries, and particularly at public facilities, among women with less education and to a lesser extent more education may be due to their role in promoting the free public health services and incentives within the NHM. Studies have found that ASHAs have generally had a strong role in encouraging women to attend public antenatal and delivery care, in part by promoting the *Janani Suraksha Yojana* incentives, and that this has occurred alongside shifting social norms in favour of facility-based maternity care in India.[44–47] Our findings seem to support evidence indicating that having at least one home visit in the third trimester may contribute to improving institutional delivery not only overall but equitably, by facilitating access to the more affordable care in the public health system in UP. It would be valuable for future research to examine whether having multiple home visits leads to even better outcomes, and to compare this for women of lower and higher age and birth order who tend to be at greater risk of perinatal loss.

This quantitative study indicated that ASHAs' home visits played the strongest role in supporting women of lower education groups to give birth at public facilities, among whom there were lower PMRs. Yet women from the lower, and even higher, education groups who delivered at private facilities still reported high perinatal mortality, though there were fewer cases leading to wider

**Table 3** Adjusted associations between perinatal mortality rate and any third trimester home visits, overall and by education and place of birth

| Third trimester home visits | All education groups (n=52 588) | | None/illiterate (n=30 600) | | 1–5 standards (n=5689) | | 6–10 standards (n=9887) | | >10 standards (n=6412) | |
|---|---|---|---|---|---|---|---|---|---|---|
| | Adjusted PMR (95% CI) | Adjusted RR (95% CI) | Adjusted PMR (95% CI) | Adjusted RR (95% CI) | Adjusted PMR (95% CI) | Adjusted RR (95% CI) | Adjusted PMR (95% CI) | Adjusted RR (95% CI) | Adjusted PMR (95% CI) | Adjusted RR (95% CI) |
| *All births, not adjusting for place of birth* | | | | | | | | | | |
| Any (n=28 686) | 41.5 (39.2 to 43.8) | Ref | 45.9 (42.7 to 49.2) | Ref | 37.8 (21.2 to 44.3) | Ref | 35.6 (30.8 to 40.4) | Ref | 33.1 (27.3 to 38.9) | Ref |
| None (n=23 902) | 48.6 (45.9 to 51.3) | 1.17 (1.08 to 1.27) | 50.7 (47.2 to 54.2) | 1.10 (1.00 to 1.22) | 56.1 (47.0 to 65.1) | 1.48 (1.17 to 1.88) | 46.3 (40.0 to 52.6) | 1.30 (1.07 to 1.58) | 36.3 (29.3 to 43.3) | 1.10 (0.84 to 1.42) |
| *All births, adjusting for place of birth* | | | | | | | | | | |
| Any (n=28 686) | 41.3 (39.0 to 43.6) | Ref | 45.5 (42.2 to 48.7) | Ref | 37.3 (30.8 to 43.8) | Ref | 36.0 (31.1 to 40.9) | Ref | 33.3 (27.5 to 39.2) | Ref |
| None (n=23 902) | 48.9 (46.1 to 51.6) | 1.18 (1.09 to 1.28) | 51.2 (47.7 to 54.8) | 1.13 (1.02 to 1.25) | 56.9 (47.7 to 66.2) | 1.53 (1.20 to 1.94) | 45.7 (39.4 to 52.0) | 1.27 (1.05 to 1.54) | 35.9 (28.9 to 42.9) | 1.07 (0.83 to 1.40) |
| *Home births* | | | | | | | | | | |
| Any (n=8295) | 37.8 (33.7 to 42.0) | Ref | 41.1 (35.8 to 46.4) | Ref | 23.7 (13.7 to 33.7) | Ref | 35.5 (25.1 to 45.9) | Ref | 30.0 (14.9 to 45.1) | Ref |
| None (n=10 935) | 39.7 (36.1 to 43.3) | 1.04 (0.91 to 1.21) | 41.6 (37.2 to 45.9) | 1.01 (0.86 to 1.19) | 35.4 (24.4 to 46.5) | 1.50 (0.88 to 2.53) | 38.0 (28.2 to 47.8) | 1.07 (0.72 to 1.58) | 26.3 (12.6 to 40.0) | 0.88 (0.42 to 1.82) |
| *Public facility births* | | | | | | | | | | |
| Any (n=17 426) | 38.4 (35.5 to 41.3) | Ref | 43.8 (39.6 to 48.0) | Ref | 35.6 (27.6 to 43.6) | Ref | 31.8 (26.2 to 37.4) | Ref | 30.4 (23.6 to 37.2) | Ref |
| None (n=10052) | 50.4 (46.2 to 54.6) | 1.31 (1.17 to 1.47) | 54.5 (48.5 to 60.4) | 1.24 (1.08 to 1.44) | 62.7 (48.3 to 73.6) | 1.76 (1.28 to 2.43) | 44.0 (35.3 to 52.8) | 1.38 (1.06 to 1.81) | 35.3 (25.7 to 44.9) | 1.16 (0.82 to 1.65) |
| *Private facility births* | | | | | | | | | | |
| Any (n=3017) | 68.5 (59.6 to 77.4) | Ref | 84.8 (69.6 to 100.0) | Ref | 96.0 (62.4 to 129.6) | Ref | 56.9 (40.0 to 73.8) | Ref | 43.5 (29.0 to 58.0) | Ref |
| None (n=2915) | 76.5 (67.0 to 86.1) | 1.12 (0.93 to 1.34) | 93.1 (76.6 to 109.5) | 1.10 (0.85 to 1.41) | 104.5 (70.8 to 138.8) | 1.09 (0.68 to 1.76) | 69.4 (50.7 to 88.0) | 1.22 (0.82 to 1.82) | 45.5 (30.8 to 60.2) | 1.05 (0.66 to 1.66) |

CI, confidence interval; PMR, perinatal mortality rate; RR, risk ratio.

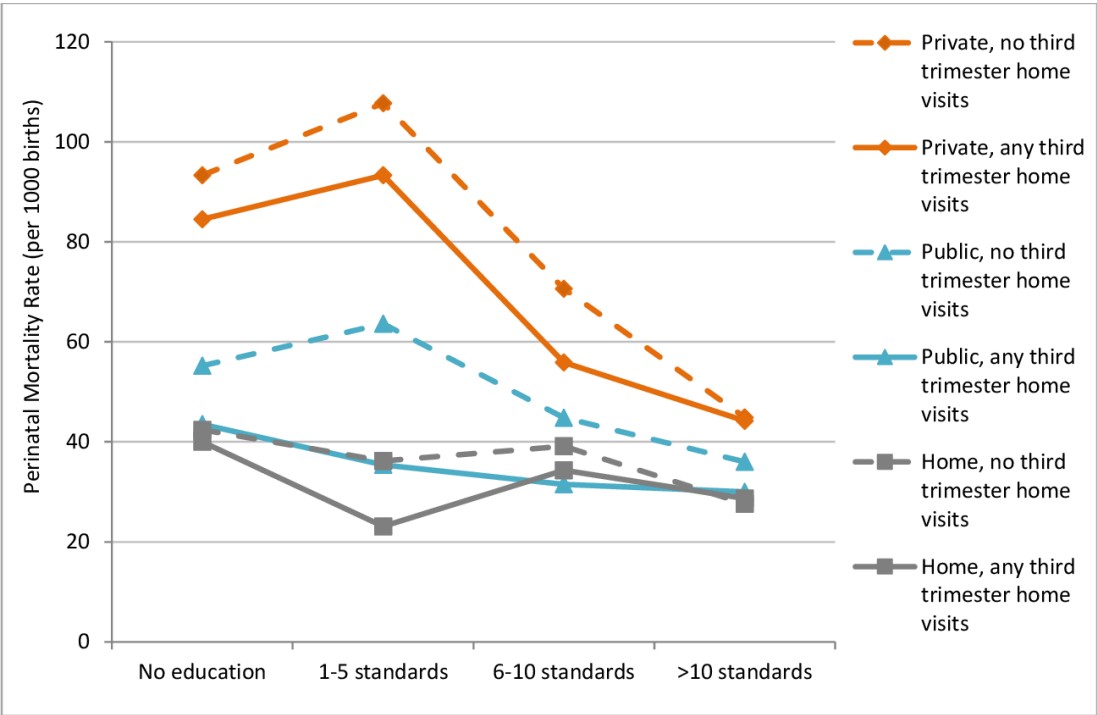

**Figure 2** Perinatal mortality rate among women in each education group, by exposure to any third trimester home visits and place of birth.

confidence intervals. Relatedly, the qualitative component of our wider mixed-methods study suggested that ASHAs' home visits encouraged more poorer and less educated women to deliver at public community health centres.[48] Those who had to be referred to higher-level, often private facilities, to receive emergency obstetric care faced delays and economic barriers that increased their risk of perinatal loss and thereby perpetuated inequities. It is therefore important for programmes to consider that improvements in institutional delivery have not always translated into reductions in mortality among mothers and their babies where there has been lower quality of services and poor referral mechanisms, particularly at primary healthcare facilities or unregulated private services where people of lower socioeconomic positions often attend first.[49 50] More research is needed to understand the processes by which inequitably high PMRs continue for women of lower socioeconomic positions, especially those who face pregnancy and intrapartum complications, in different contexts.

In 2018, the Government of India launched the *Ayushman Bharat* Programme to enhance the availability of free integrated primary healthcare services through village-level Health and Wellness Centers.[51] The programme aims to attach five ASHAs to each centre, where they will work with new mid-level health providers. ASHAs are to continue home visits to promote utilisation and safe health behaviours for RMNCH, among a range of other health services.[51] A systematic review of previous studies in India and elsewhere found that lower neonatal mortality was associated with CHW home visits, particularly when they included both preventive (eg, counselling

on care seeking or behaviours) and curative roles (eg, provision of injectable antibiotics).[23] Another showed that the effects of CHW home visits on care-seeking for institutional delivery can be equal or greater for lower compared with higher wealth or education groups.[52] It will be crucial to support ASHAs to continue providing counselling on institutional delivery, essential newborn care, as well as timely identification, treatment or referrals for women that also face greater vulnerability due to socio-economic disadvantage.[3 51 53 54]

Building on the NHM, the *Ayushman Bharat* policies are also explicitly based on the premise that the health system functions effectively.[51] The programme includes public as well as empanelled private first referral units to be covered by the insurance scheme called *Pradhan Mantri Jan Arogya Yojana*, which should remove financial barriers for women of lower socioeconomic groups facing birth complications.[55] As more women and their families are giving birth at public and private health facilities, it is particularly important to consider the pluralistic health system in India holistically when addressing maternal and perinatal health inequities.[50 55–57] This would necessitate targeting the causes of differential gaps in accessibility, quality and integration of the private and public health sectors, and between community and facility-based services, so as to strategically address persistent inequalities in perinatal mortality.

## CONCLUSIONS

This study's findings suggested that ASHAs' contacts and counselling for pregnant women through home

visits in the third trimester may have influenced them to give birth at a public facility, and particularly those with lower compared with higher education levels. Further, less educated women who had any compared with no home visits that delivered at a public facility appeared to have the lowest rates of perinatal mortality. Future research, policies and programmes should seek to further understand the processes by which CHWs' home visits can contribute to improving equity in perinatal health outcomes, and how more integrated community and health system efforts can strategically support these efforts.

**Author affiliations**
[1]Institute for Global Public Health, Department of Community Health Sciences, Rady Faculty of Health Sciences, University of Manitoba, Winnipeg, Manitoba, Canada
[2]Institute for Global Health, Faculty of Population Health Sciences, University College London, London, UK
[3]India Health Action Trust, Lucknow, India
[4]Department of Public Health, Erasmus Medical Center, Rotterdam, The Netherlands

**Contributors** AKB planned and conducted the analyses, and wrote the draft manuscript. TAH, TC and AP provided guidance on the design of the study and support throughout the analyses. BMR, SI, JA and BD designed the CBTS methodology, managed the data collection and quality assurance, and gave technical input on the analyses. All authors read and gave input on the manuscript.

**Funding** This work was supported by the Bill & Melinda Gates Foundation (OPP1161429). AKB's time was funded by University College London's Overseas and Graduate Research Scholarships. TAJH's time was funded by a Research Excellence Initiative grant from Erasmus University Rotterdam, The Netherlands. The funders had no role in the design of the study, data collection and analysis, planning or writing of the manuscript.

**Competing interests** None declared.

**Patient consent for publication** Not required.

**Ethics approval** Data collection protocols and analysis plans were approved by the Institutional Review Board of Sigma Research and Consulting in New Delhi (#10040/IRB/D/16-17) and the University College London Research Ethics Committee (#9909/001).

**Provenance and peer review** Not commissioned; externally peer reviewed.

**Data availability statement** Data are available upon reasonable request. The data are hosted on the University of Manitoba server, and a de-identified data cut along with the questionnaires and data dictionary may be made available upon reasonable request (contact email: Ramesh.BanadakoppaManjappa@Umanitoba. ca).

**ORCID iD**
Andrea Katryn Blanchard http://orcid.org/0000-0002-5118-8942

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
