## [Reviewer comments · BMJ Open]

ARTICLE DETAILS

TITLE (PROVISIONAL)	Associations between community health workers' home visits and education-based inequalities in institutional delivery and perinatal mortality in rural Uttar Pradesh, India: a cross-sectional study
AUTHORS	Blanchard, Andrea; Colbourn, Timothy; Prost, Audrey; Ramesh, Banadakoppa Manjappa; Isac, Shajy; Anthony, John; Dehuri, Bidyadhar; Houweling, TA

VERSION 1 – REVIEW

REVIEWER	Bashar, MD PGIMER, COMMUNITY MEDICINE
REVIEW RETURNED	18-Dec-2020

GENERAL COMMENTS	1. The study addresses an important aspect of effect of ASHA's visit on education based inequalities in perinatal health. The study objectives and methodology are clearly stated with sound background.2. The abstract requires mention of statistical analyses and the outcome measures and how they were performed.3. Detailed of field investigators EDUCATIONAL BACKGROUND AND training received is required. Whether it was hands on training or didactic lecture based?4. Language & Grammar of the manuscript requires substantial improvement particularly in use of SYNTAX
---

REVIEWER	Hanson, Claudia Karolinska Institute, Dept Public Global Health
REVIEW RETURNED	01-Jan-2021

GENERAL COMMENTS	I think this is a really nice and good paper, but please check regarding an ethics statement, I might have overlooked, but could not find.
--

REVIEWER	Tsegaye, Berhan Hawassa University, Midwifery
REVIEW RETURNED	11-Jan-2021

GENERAL COMMENTS	Reviewer comment Title: The effects of community health workers' home visits on education-based inequalities in perinatal health in rural Uttar Pradesh, India General comment I want to thank the editor to give the chance to review this paper as it is related with my previous work as well as my profession. Generally, the current study is the secondary analysis of data collected some years back. It has many strengths which enable it for
--

	publication as well as its weakness. For example, it uses large sample size (as most mortality rate studies use large sample size for this rare outcome) and cover many states. Moreover, the study utilized the standard tool for data collection. All of these condition can lead for better data for conclusion about target population. However, I have many concerns and questions as well as suggestions which must be resolved before going to the next step.  1. The second objective of the manuscript (the effect of home visit on institutional delivery) is not reflected in the title? The title should be in line with objective. 2. From the title the reader may guess the design is longitudinal (follow up) study. But, it is cross section. Hence, better to avoid effect or add design in the title. For example, 'community workers home visits are associated with institutional delivery and perinatal death' 3. The author should avoid abbreviation in the abstract 4. The fact that the large sample size makes the finding more reliable. The author also described this as strength but the very large nature of sample size has also its problem. For example, the very narrow effect size in the result was associated with extremely large sample size. If the sample size is very large it could detect small association easily which are not significant in fact. So, I suggest this was happened in this paper by simply observing the width of confidence interval (effect size). Therefore, the sample size should be not very small as the same time not very large. 5. How PSU are selected, what are the SSU (secondary sampling units) and How they were selected? 6. How study participants were allocated in each areas(clusters)? Was there sampling frame? 7. We understood the sampling design is multi-stage with two stage? why the author did not use design effect for sample size calculation as assumption? 8. The exclusion criteria is incorrect from the very beginning women who abort were nor the study population as birth and abortion are different? 9. The author should explain who he/she did for households more than 1 eligible woman? 10. The author should explain the study in detail. Is that institutional based of community based? The study seems community based study, if so what was the sampling unity and study unit? 11. If two eligible women exist in the same household by which method the author, select? 12. The author should include the 80% of nonresponse rate of the study as limitation. Since it reduced the generalizability (applicability) or the external validity of the result. 13. I am not clear about the sample size assumption. If there are 100 blocks in the 25 areas, why the author took 110(where were extra 10?). 14. Furthermore, why the author takes 15% non-response assumption. We know the title is not sensitive in which much non response rate is not expected. The other assumption of 7.5% expected change was not clear? I need clear explanation on the sample size estimation. Generally, the authors try to mention the limitation, adequately discuss and recommend based on the result appropriately. I recommend the major revision if the authors can address the issues above.
--	--

VERSION 1 – AUTHOR RESPONSE

REVIEWER: 1

Dr. MD Bashar, PGIMER

Comments to the Author:

1. Reviewer 1: The study addresses an important aspect of effect of ASHA's visit on education based inequalities in perinatal health. The study objectives and methodology are clearly stated with sound background.
2. Reviewer 1: The abstract requires mention of statistical analyses and the outcome measures and how they were performed.

Authors: We have included that the analyses involved generalised linear modelling under lines 3-4 in the Methods in the Abstract (p.2), and have now specified the associations between exposure and outcome measures that we examined.

3. Reviewer 1: Detailed of field investigators EDUCATIONAL BACKGROUND AND training received is required. Whether it was hands on training or didactic lecture based?

Authors: The educational background for field investigators was a minimum of a Bachelor degree in social science or related field, and 6 months or more of related experience. The training they received was first didactic orientation and then hands-on field practice on which supervisors observed and gave feedback. This has been added to line 1-4, paragraph 3, p.6.

4. Reviewer 1: Language & Grammar of the manuscript requires substantial improvement particularly in use of SYNTAX

Authors: We have now aimed to improve syntax and readability by shortening or rearranging complicated sentences throughout the paper. We also ensured that there was correct grammar and use of consistent tenses.

REVIEWER: 2

Dr. Claudia Hanson, Karolinska Institute

Comments to the Author:

Reviewer 2: I think this is a really nice and good paper, but please check regarding an ethics statement, I might have overlooked, but could not find

Authors: We appreciate the reviewers' comment. An ethics statement can be found on page 9, paragraph 4. We have added the number of the ethics approval as well.

REVIEWER: 3

Dr. Berhan Tsegaye, Hawassa University

Comments to the Author:

Reviewer comment

Title: The effects of community health workers' home visits on education-based inequalities in perinatal health in rural Uttar Pradesh, India

General comment I want to thank the editor to give the chance to review this paper as it is related with my previous work as well as my profession. Generally, the current study is the secondary analysis of data collected some years back. It has many strengths which enable it for publication as well as its weakness. For example, it uses large sample size (as most mortality rate studies use large sample size for this rare outcome) and cover many states. Moreover, the study utilized the standard tool for

data collection. All of these condition can lead for better data for conclusion about target population. However, I have many concerns and questions as well as suggestions which must be resolved before going to the next step.

1. Reviewer 3: The second objective of the manuscript (the effect of home visit on institutional delivery) is not reflected in the title? The title should be in line with objective.

Authors: We have now specified the two outcomes in the title, including institutional delivery and perinatal mortality.

2. Reviewer 3: From the title the reader may guess the design is longitudinal (follow up) study. But, it is cross section. Hence, better to avoid effect or add design in the title. For example, 'community workers home visits are associated with institutional delivery and perinatal death'

Authors: We have now modified the language of the title to specify that the analyses are cross-sectional and therefore looked at associations between the community health workers' home visits and lower education-based inequalities in the outcomes of institutional delivery and perinatal mortality.

3. Reviewer 3: The author should avoid abbreviation in the abstract

Authors: We have now removed abbreviations for perinatal mortality, but left ASHAs and Rate Ratios (RR), which are well-known acronyms. This allowed it to stay within the word length while retaining more on the content of the paper in the abstract.

4. Reviewer 3: The fact that the large sample size makes the finding more reliable. The author also described this as strength but the very large nature of sample size has also its problem. For example, the very narrow effect size in the result was associated with extremely large sample size. If the sample size is very large it could detect small association easily which are not significant in fact. So, I suggest this was happened in this paper by simply observing the width of confidence interval (effect size). Therefore, the sample size should be not very small as the same time not very large.

Authors: We agree that the large sample size was valuable for detecting associations, while also making it harder to assess the relative significance in some of the comparisons. Our rationale for using this large sample was indeed because it allowed us to look not only at associations between home visits with inequalities in institutional delivery between education groups, which is possible with a smaller sample, but also to look at this for perinatal mortality as a rarer event. We found that this large sample was needed to stratify the PMR analyses by education and especially place of delivery (and even some of those confidence intervals were wider), allowing us to show that the associations between exposure to ASHAs' home visits and PMR were weaker and less equitable among deliveries at the private compared to public facilities.

5. Reviewer 3: How PSU are selected, what are the SSU (secondary sampling units) and How they were selected?

Authors: The PSUs were ASHA areas, and they were selected through the systematic random sampling method within each study block. All households were eligible within the PSU. This is described under sampling and participants on page 6 (paragraph 1, lines 6-7 and paragraph 2, lines 1-2).

6. Reviewer 3: How study participants were allocated in each areas(clusters)? Was there sampling frame?

Authors: We used a sampling frame for the selection of ASHA areas as PSUs, by listing them all and selecting 110 per block using systematic random sampling as mentioned above. For the selection of eligible participants within the PSUs, all households were screened for women who completed pregnancies within the reference period of up to 60 days, and all eligible women were invited for an interview.

7. Reviewer 3: We understood the sampling design is multi-stage with two stage? why the author did not use design effect for sample size calculation as assumption?

Authors: It is correct that the CBTS used a two-stage sampling design, which is now indicated on page 5, paragraph 3, lines 1-2. The study's design effect was deviation from simple random sampling, and it was assumed to be 1.5 (added to line 6). The reason was that the most recent NFHS in Uttar Pradesh took the estimated design effect for institutional delivery in rural areas to be 1.76. Since the CBTS included PSUs within a block selected all at one stage, and all eligible respondents within the PSU were selected, the design effect was assumed to be slightly smaller at 1.5. The calculation is below for further clarification.

The sample size is estimated using the following formula:

Where...

n = desired sample size

D= design effect (assumed to be 1.5)

P1= the estimated level of indicator at round 1 (based on AHS 2011-12)

P2= the expected level of indicator at subsequent round

$P = (P1+P2)/2$

$Z(1-\alpha) = Z$ score corresponding to the probability which is desired to be able to conclude that an observed change in indicator ($P2-P1$) occurred by chance or due to random fluctuation of sampling (fixed at 0.05)

$Z(1-\beta) = Z$ score corresponding to the degree of confidence with which it is desired to be certain of detecting a change in indicator ($P2-P1$) if one actually occurred (fixed at 0.8)

8. Reviewer 3: The exclusion criteria is incorrect from the very beginning women who abort were nor the study population as birth and abortion are different?

Authors: We have removed this word 'excluding' women who had an abortion to avoid the confusion that this was exclusion criteria for participants from within the sample for the present study. We also amended the sentence about eligible participants to indicate that they were any women who completed a pregnancy (not only a birth, but also pregnancies that ended in any kind of abortion) within the last 0-60 days (p.6, paragraph 2, lines 6-7). In this way, those who had an abortion were not part of the sample included for our analyses.

9. Reviewer 3: The author should explain who he/she did for households more than 1 eligible woman?

Authors: All eligible women were included, even if they were in the same household, which we have now added on page 6, paragraph 2, line 1.

10. Reviewer 3: The author should explain the study in detail. Is that institutional based of community based? The study seems community based study, if so what was the sampling unity and study unit?

Authors: We have included that the survey was a cross-sectional survey, and now added it was community-based on page 5, paragraph 2, line 8. We included that the sampling unit was an ASHA area on p.5, paragraph 3, line 3.

11. Reviewer 3: If two eligible women exist in the same household by which method the author, select?

Authors: As mentioned above, all eligible women were invited even if they were in the same household.

12. Reviewer 3: The author should include the 80% of nonresponse rate of the study as limitation. Since it reduced the generalizability (applicability) or the external validity of the result.

Authors: We have now added that the response rate of 80% was a limitation on p.15, paragraph 2, lines 3-4. This response rate of 80% (non-response of 20%) mostly resulted from women not being available in their home because they were still at the facility or had returned to their maternal home within the 0-60 days post-delivery period of eligibility. However, this time period was chosen in part because it would mitigate against recall bias.

13. Reviewer 3: I am not clear about the sample size assumption. If there are 100 blocks in the 25 areas, why the author took 110(where were extra 10?).

Authors: We realize this sentence was not clear, and have reworded to indicate that 110 PSUs (ASHA areas), rather than blocks, were chosen through systematic random sampling among a list of all ASHA areas in the 100 blocks of 25 districts (page 6, paragraph 1, lines 7-8).

14. Reviewer 3: Furthermore, why the author takes 15% non-response assumption. We know the title is not sensitive in which much non response rate is not expected. The other assumption of 7.5% expected change was not clear? I need clear explanation on the sample size estimation.

Authors: We have now tried to clarify the sample size estimation on p.5, paragraph 3 to p.6, paragraph 1. We aimed to indicate that the sample size per block that would be required to detect a minimum percentage change of 7.5% within six months for the key indicator of institutional delivery was ideal (n=536 per ASHA area needed), because at a lower minimum detectable change of 5% it would require too large a sample for the available resources (n=1209 per block), and to detect a 10% change it would be unnecessarily small (n=300). An assumed 15% non-response rate was chosen because we expected there may be a lower response rate particularly among women who were still at the health facility just after delivery, or those who went to their maternal home for the delivery within the eligibility period of 0-60 days after giving birth.

VERSION 2 – REVIEW

REVIEWER	Bashar, MD PGIMER, COMMUNITY MEDICINE
REVIEW RETURNED	20-Mar-2021
GENERAL COMMENTS	The ARTICLE HAS BEEN REVISED AS PER THE COMMENTS GIVEN.. However, There is no mention regarding the consent taken from the patients
REVIEWER	Svensson, Elisabeth Orebro University
REVIEW RETURNED	26-Apr-2021
GENERAL COMMENTS	Bmjopen-2020-044835.R1

	Associations between community health workers' home visits and lower education-based inequalities in institutional delivery and perinatal mortality in rural Uttar Pradesh, India I have been asked to review this revised manuscript which reports results of a comprehensive population-based study regarding possible impacts of health workers' home visit on perinatal health inequalities. Being a statistical and methodological expert in scientific problem solving, I found this topic interesting. This study is based on an extensive data material that offers good conditions for valuable results – provided a correct data management, appropriate statistics and research methods. However, there are major concerns regarding statistical methods and the lack of compliance to the basic rules of research methodology, and how to write a scientific paper! The INRODUCTION describes well why this study is important. However, the AIM should be specified with research questions including the variables of interest. The METHODS sections should describe how the study was planned to be performed. The design, sampling and data collection are well written. However, parts of the information given in the variables sections seem to refer to, or belong to statistical methods and to results.  - Page 39 line 26: Do not start a sentence with a number: "5,173 participants responded..." - A large number of acronyms are used. PMR on page 40 line 5 should be defined on page 39 line 53 in connection with "perinatal mortality rates". - Page 40 Independent variables: line 30: it is good that the authors mention that the variable "educational level" was operationally defined by four ordered categorical levels, which implies ordinal data. Which other variables were registered as ordered categorical (ordinal) data? Did the statistical methods take account of the measurement properties of ordinal data? - Page 40 line 43: "We considered exposure to home visits earlier in pregnancy, but found that only 9.8% of women received a home visit ..." How was this information found? The source of information should be mentioned. If this is a result of this reported study then it is a result! - Page 41 lines 6-12: "...Among these, the available covariates in the dataset that were found to be significant at $p < 0.05$ in bivariable models were retained in multivariable models." If this is a result of this reported study, the information should be moved to statistical methods and/or to the results sections. - Page 41 lines 12-28: please check the types of information. It seems that there is a mixture of statistical methods and results together with the variables.
--	---

STATISTICAL METHODS

The statistical methods for description and evaluation should be linked to the study aim and to specified research questions and to the variables of interest. The links between research questions (What?), the variables and the planned statistical methods (How?) are completely missing.

Statistical methods must be planned before looking at the data, and must not be integrated with the data, and results. There are serious signs of “fishing” and “data dredging” in this section.

The statistical methods must be described sufficiently regarding each of the research questions/aims to allow theoretical replications of results, given the raw data! This section does not meet the basic requirements of a Statistical methods section in a scientific paper.

In the Abstract the concept “cross-sectional study” was used, which means that this is an observational study that cannot admit causative conclusion, but can report association, and describe independent group comparisons. Which variables, which groups, which research questions were planned to be described, compared – and how?

There is a serious mixture of results and evaluation plans in this section.

- Was all the data management, the calculations (data descriptions) as well as the data analyses performed by the STATA?

- The kind of measures that were used to describe the participants and the variables of interest: frequencies, proportions, mortality rate, mean, standard-deviation, median, quartiles etc should also be mentioned?

- Table 1 is a statistical description of the participants, and is a result and should NOT be presented in Statistical methods!

- Furthermore, Table 1 is NOT a cross-table.

- Tables should not have vertical lines.

- Figures 1, 2 are good graphical descriptions of the participants, which means RESULTS and should NOT be presented in methods section!

- Is it so that the supplementary figures could be the authors' models for how to evaluate the associations? Then it would be good to use the figures to motivate the design and the planned statistical methods of description, comparisons and associations.

RESULTS

The results of the research questions shall be given without using subjective expressions like “the majority”, “over 80%” “almost” etc.

	In comparative studies confidence intervals for describing the difference in proportions between the groups should be used. In case of a large number of comparisons, adjustments for multiplicity should be used. Because of the lack of compliance to the basic rules of research methodology, and how to write a scientific paper the results cannot be commented further.
--	---

VERSION 2 – AUTHOR RESPONSE

Reviewer: 1

REVIEWER 1: The ARTICLE HAS BEEN REVISED AS PER THE COMMENTS GIVEN.. However, There is no mention regarding the consent taken from the patients

AUTHORS’ RESPONSE: We appreciate the reviewers’ time in confirming that we have addressed all their comments. The manuscript includes that informed consent was taken by participants (Methods, page 7, paragraph 2, line 7 under the section on Data collection).

Reviewer: 4

Authors: We appreciate the Reviewer’s time in providing detailed comments on our paper. We have responded to related Reviewer comments in three sections, including comments to clarify how the aim, questions, methods and variables are related; comments on the distinction between methods and results; and additional comments on style and clarification.

1) **Comments on the aim of the study in relation to research questions, methods and variables**

REVIEWER 4: The INTRODUCTION describes well why this study is important. However, the AIM should be specified with research questions including the variables of interest.

In the Abstract the concept “cross-sectional study” was used, which means that this is an observational study that cannot admit causative conclusion, but can report association, and describe independent group comparisons. Which variables, which groups, which research questions were planned to be described, compared – and how?

The statistical methods for description and evaluation should be linked to the study aim and to specified research questions and to the variables of interest. The links between research questions (What?), the variables and the planned statistical methods (How?) are completely missing.

The statistical methods must be described sufficiently regarding each of the research questions/aims to allow theoretical replications of results, given the raw data! This section does not meet the basic requirements of a Statistical methods section in a scientific paper.

- Is it so that the supplementary figures could be the authors’ models for how to evaluate the associations? Then it would be good to use the figures to motivate the design and the planned statistical methods of description, comparisons and associations.

AUTHORS’ RESPONSE: We have stated the aim of our study at the end of the Background section, in a format that we feel is appropriate for the field of global public health. Our aim was to understand whether receiving community health workers’ home visits was more greatly

associated with higher institutional delivery and lower perinatal mortality rates among women who were less compared to more educated. The aim and focus of the study's analyses was consistent from the beginning based on gaps in the literature and the aims of India's National Health Mission programme for improving health inequities. We completely agree and have acknowledged that this study does not imply or speak to causal relationships, but we feel it does point to whether there are associations that can be explored more qualitatively, as the authors have also done within the broader mixed methods study, as well as quantitatively in other settings.

We think that the reviewer has identified an important issue in these comments that can be addressed by being more explicit in laying out the research questions and sub-questions, and how they were operationalized through each step of statistical methods and the variables used. The two conceptual models in Supplementary figures 1 and 2 were intended to guide these steps as the Reviewer suggested. We have now added in the Background that to fulfil this study's aim, our analyses were guided by three research questions, analyzed first for institutional delivery and then for perinatal mortality:

- 1) Do women with higher compared to lower education levels have a) lower institutional delivery, and b) lower perinatal mortality rates, overall and at home, private or public facility births? This was to establish whether education-based inequalities in the outcomes exist.
- 2) Is women's exposure to any third trimester home visits associated with a) higher institutional delivery, and b) lower PMR, overall and by place of birth? This was to establish whether there is any association between the exposure and the outcomes.
- 3) Is exposure to third trimester home visits more strongly associated with a) higher institutional delivery, and b) lower perinatal mortality, for women with lower compared to higher education levels overall and at each place of birth? If for the first two questions, the results show education-based inequalities exist overall and home visits are associated with the outcomes, the final question addresses the overall aim of the study to understand whether having any home visits is as much or more greatly associated with the outcomes among women in lower compared to higher education groups (indicating inequalities in the outcomes).

We now refer to the Supplementary figures with our conceptual models in the Background section, instead of in the Methods section as before. We have also aimed to explain how these questions were answered in turn using the statistical analyses within the Methods on pages 9-10. In the Results, we now also indicate where the three research questions are each addressed for institutional delivery, and then perinatal mortality on pages 12-13 and 15 respectively.

REVIEWER 4: In comparative studies confidence intervals for describing the difference in proportions between the groups should be used. In case of a large number of comparisons, adjustments for multiplicity should be used.

AUTHORS' RESPONSE: We feel that this comment stems from the assumption that we have introduced multiple comparisons *ad hoc*, and that the models assess multiple variables at the same time. Instead, the comparisons we made were conceptually driven, and the variables were not introduced all at one time, so the issue of multiplicity would be less of a concern. For example, we look at the association between home visits and perinatal mortality at three delivery places in separate models. An analysis and adjustment for multiplicity would also be more relevant if we included a wide range of independent variables in the model at one time to identify which were significantly related to the outcomes, potentially resulting in a greater likelihood of type I error. However, we focused on one exposure variable (home visits), and introduced conceptually-relevant covariates in the dataset separately in the bivariate analyses first, and then only included those in the final multivariate regression models that were significant at $p < 0.05$ as is common practice (addressed in related comment below). We were also not interested in each of their associations with the outcomes in the multivariable models.

Furthermore, the comparisons between education groups were within a single variable rather than separate ones so multiplicity would also not apply to those comparisons. We were most interested in the relative risk and absolute adjusted PMR between women in four education categories rather than testing significance between education categories. It would not be very meaningful to apply a

correction for multiplicity because we are not comparing the significance using p-values but rather the actual adjusted RR and PMR values.

2) Comments regarding the distinction between methods and results

REVIEWER 4: The METHODS sections should describe how the study was planned to be performed. The design, sampling and data collection are well written. However, parts of the information given in the variables sections seem to refer to, or belong to statistical methods and to results.

- Page 40 line 43: “We considered exposure to home visits earlier in pregnancy, but found that only 9.8% of women received a home visit ...” How was this information found? The source of information should be mentioned. If this is a result of this reported study then it is a result!
- Page 41 lines 6-12: “...Among these, the available covariates in the dataset that were found to be significant at $p < 0.05$ in bivariable models were retained in multivariable models.” If this is a result of this reported study, the information should be moved to statistical methods and/or to the results sections.
- Page 41 lines 12-28: please check the types of information. It seems that there is a mixture of statistical methods and results together with the variables

Statistical methods must be planned before looking at the data, and must not be integrated with the data, and results. There are serious signs of “fishing” and “data dredging” in this section.

- Table 1 is a statistical description of the participants, and is a result and should NOT be presented in Statistical methods!
- Figures 1, 2 are good graphical descriptions of the participants, which means RESULTS and should NOT be presented in methods section!

AUTHORS' RESPONSE: This set of comments relates to the first, and we would like to reiterate that our analytical methods were conceptualized and planned before they were conducted. As described early in the paper, the study's aim built from our assessment of gaps in existing research and the aims of the National Health Mission's ASHA programme for improving perinatal health equity. In addition, we strongly feel that the reviewers' suggestion that there is a conflation of methods and results in the Methods section does not align with well-accepted statistical approaches in this type of cross-sectional, population-based study. After identifying which exposure and outcome variables we were interested in analyzing *a priori* based on the literature and programme context, we necessarily explored the data first to identify the most appropriate way to analyze them (e.g. as binary or ordinal and with which categories), and to know which covariates and confounding variables should be included to ensure parsimonious but also theoretically and statistically robust multivariable models. There are three specific points raised by the reviewer to clarify in this regard:

First, relating to the measure of exposure to home visits, we considered the distribution of home visits between different trimesters of pregnancy. When exploring the frequencies of women having home visits in each trimester in the CBTS dataset, we saw that most women who had any visits in the first and second trimester also had one in the third (i.e. as stated in the paper, only 9.8% had one in the third but not earlier). Therefore, we decided to look at a binary exposure variable of women having any third trimester home visits, which helps the analytical comparisons to be clearer and is also more relevant programmatically because ASHAs counsel more on safe birth and newborn care practices that would prevent perinatal mortality (we added details on this in response to the first set of reviewers). This is not a result in relation to our research questions, but a necessary methodological step to prepare for the main analysis.

The second point for clarification is regarding the inclusion of covariate and confounding variables. It is well-accepted that when conducting regression analyses in population health sciences, we would start with running bivariable (or bivariate) models with each potentially relevant variable that is available in the data (selected based on literature and context, as described on page 8). Then only those that were identified as significant at $p < 0.05$ are kept in the final models to remove the potential for collinearity between covariates themselves. Therefore, we think that this approach and related explanation within our Statistical analyses section is appropriate rather than reporting this as a result.

Finally, we see that there is a misunderstanding when the reviewer says that we should not include our Table 1, Figure 1-2 and so on within the Methods. While we had indicated which parts of the analysis were related to which figures/tables in the Methods section, the actual figures/tables are presented solely within the Results. We have removed any reference to the tables or figures from within the Methods to avoid that confusion.

3) Additional comments on style and clarification

REVIEWER 4: Page 39 line 26: Do not start a sentence with a number: “5,173 participants responded...”

AUTHORS' RESPONSE: We have now removed sentences starting with numbers as suggested.

REVIEWER 4: A large number of acronyms are used. PMR on page 40 line 5 should be defined on page 39 line 53 in connection with “perinatal mortality rates”.

AUTHORS' RESPONSE: We realize that acronyms can reduce readability, and have made sure to introduce all acronyms at first use, and removed some (e.g. UP TSU, GLM) where we did not use them often.

REVIEWER 4: Page 40 Independent variables: line 30: it is good that the authors mention that the variable “educational level” was operationally defined by four ordered categorical levels, which implies ordinal data. Which other variables were registered as ordered categorical (ordinal) data? Did the statistical methods take account of the measurement properties of ordinal data?

AUTHORS' RESPONSE: We have now added a note for all variables whether it was binary or categorical, and if the latter, whether categories were ordinal or nominal under the Variables section on pages 7-8. We also added in the Statistical analyses section that we incorporated whether a variable was ordinal within the regression analyses commands in STATA, which takes account of this and generates separate estimates for each category in relation to the reference category on page 10, paragraph 2, lines 6-7.

REVIEWER 4: Was all the data management, the calculations (data descriptions) as well as the data analyses performed by the STATA?

AUTHORS' RESPONSE: All data management, descriptive analyses and regression analyses were conducted using STATA software, which is stated on page 9, paragraph 2, line 1.

REVIEWER 4: The kind of measures that were used to describe the participants and the variables of interest: frequencies, proportions, mortality rate, mean, standard-deviation, median, quartiles etc should also be mentioned?

AUTHORS' RESPONSE: The measures to describe the variables of interest were implied in the study, but we have now aimed to describe these measures (proportions, mortality rate) within the methods and the results on page 9, paragraph 3.

REVIEWER 4: Furthermore, Table 1 is NOT a cross-table.

AUTHORS' RESPONSE: We agree, what we meant in the paper was that Table 1 was based on cross-tabulations run to find the proportions (or rates for PMR) in each category of interest using STATA.

REVIEWER 4: Tables should not have vertical lines.

AUTHORS' RESPONSE: We have removed the lines in Table 1.

REVIEWER 4: The results of the research questions shall be given without using subjective expressions like “the majority”, “over 80%” “almost” etc.

AUTHORS’ RESPONSE: We think that these approximations are frequently used in this type of paper, as a way to improve readability by summarizing and removing redundancy with the numbers in the Table below. At the same time, we have added the actual numbers in brackets on pages 11-12 to address this.

VERSION 3 – REVIEW

REVIEWER	Svensson, Elisabeth Orebro University
REVIEW RETURNED	14-Jun-2021
GENERAL COMMENTS	The author have responded to my comments and concerns satisfactory, and the manuscript has been revised accordingly. This high-quality research, scientifically correctly presented, could be a valuable model for future researchers and reports.